# A blind benchmark of analysis tools to infer kinetic rate constants from single-molecule FRET trajectories

Markus Götz [1,20] ✉, Anders Barth [2,21], Søren S.-R. Bohr[3,4], Richard Börner [5,22], Jixin Chen [6], Thorben Cordes[7], Dorothy A. Erie[8,9], Christian Gebhardt[7], Mélodie C. A. S. Hadzic[5], George L. Hamilton [10,23], Nikos S. Hatzakis [3,4], Thorsten Hugel [11,12], Lydia Kisley [13,14], Don C. Lamb [15], Carlos de Lannoy [16], Chelsea Mahn[17], Dushani Dunukara [13], Dick de Ridder [16], Hugo Sanabria [10], Julia Schimpf[11,18], Claus A. M. Seidel [2], Roland K. O. Sigel [5], Magnus Berg Sletfjerding [3,4], Johannes Thomsen[3,4], Leonie Vollmar[11,18], Simon Wanninger[15], Keith R. Weninger[17], Pengning Xu[17] & Sonja Schmid [19] ✉

Single-molecule FRET (smFRET) is a versatile technique to study the dynamics and function of biomolecules since it makes nanoscale movements detectable as fluorescence signals. The powerful ability to infer quantitative kinetic information from smFRET data is, however, complicated by experimental limitations. Diverse analysis tools have been developed to overcome these hurdles but a systematic comparison is lacking. Here, we report the results of a blind benchmark study assessing eleven analysis tools used to infer kinetic rate constants from smFRET trajectories. We test them against simulated and experimental data containing the most prominent difficulties encountered in analyzing smFRET experiments: different noise levels, varied model complexity, non-equilibrium dynamics, and kinetic heterogeneity. Our results highlight the current strengths and limitations in inferring kinetic information from smFRET trajectories. In addition, we formulate concrete recommendations and identify key targets for future developments, aimed to advance our understanding of biomolecular dynamics through quantitative experiment-derived models.

How does biomolecular function arise from structural dynamics? This largely unsolved question is central for the understanding of life at the molecular scale. However, the transitions between various conformational states have remained challenging to detect, quantify, and interpret. Over the past two decades, single-molecule Förster resonance energy transfer (smFRET) detection has emerged as a powerful technique to study the dynamics of single biomolecules under physiological conditions using fluorescence as a readout[1]. A unique aspect of smFRET is its ability to link space and time, i.e., to connect structural with kinetic information under both equilibrium and non-equilibrium conditions, which is often unachievable using ensemble methods. By measuring the distance-dependent energy transfer from a donor to an acceptor fluorophore, distances in the range of 4 to 12 nm can be measured with sub-nanometer precision and accuracy[2]. Various experimental implementations exist that allow one to measure smFRET on diverse timescales from picoseconds to hours. All of this makes smFRET an ideal tool in the growing field of dynamic structural biology[3].

To study conformational dynamics of one single molecule for an extended time (seconds to minutes), dye-labeled biomolecules are

most commonly immobilized on passivated glass slides and imaged using camera-based brightfield detection, or confocal detection using avalanche photodiodes (APDs)[2]. The resulting fluorescence time traces have a time resolution of about 10 ms – 100 ms for the most common camera-based studies, and picoseconds for single-photon counting APDs. The observation time per single molecule is limited by photobleaching, leading to an average bandwidth of less than three orders of magnitude in time[4–6]. Conformational transitions of the biomolecule change the inter-dye distance leading to discrete steps in the fluorescence signal and the FRET efficiency (Fig. 1). This desired time-resolved distance information is convoluted with largely Gaussian noise in the experiment (from autofluorescence background, detector noise, laser fluctuations, etc.). Moreover, noise and photobleaching are intrinsically coupled: increasing the laser power for a better signal-to-noise ratio causes faster photobleaching, which reduces the temporal bandwidth of the experiment. As a result, signal interpretation in terms of biomolecular states and specific transitions between them is not trivial.

A multitude of analytical approaches have been developed to infer the number of functional states and quantify kinetic rate constants from noisy experimental data. Frequently, hidden Markov models (HMMs)[7] are used to infer an idealized state sequence from which dwell-time distributions are compiled, which are then fit (with exponentials) to obtain kinetic rate constants[8,9]. Alternatively, the transition matrix that is part of every HMM can directly be converted to kinetic rate constants. The HMM formalism is based on a discrete memoryless Markov process that infers a set of parameters (probabilities of states, transitions, and observations) to describe the observed sequence of FRET efficiencies. Many extensions of the HMM formalism have been developed[10–15] including Bayesian approaches[16–19], and very fast kinetics (low energy barrier crossings) can be inferred from single-photon arrival times[20–22].

Often, multiple input models are compared based on a scarcity criterion to avoid bias in the selection of the optimal model size (i.e., the number of states and rate constants), and hence the number of free parameters[8,23–25]. Other analysis approaches, such as correlation analysis[26–30] and discretization methods based on cluster analysis[31–34], may treat the raw data in a model-free way while the extraction of individual rate constants (rather than residence times only) still relies on an initial guess of a model. The growing number of analytical methods renders it increasingly difficult to decide on the optimal tool

for a specific application and to judge whether the described improvements justify the time cost of implementation. Hence, it was identified during a round table discussion of the smFRET community (Fluorescence subgroup, Biophysical Society Meeting 2019, Baltimore, US) that a critical assessment of the available tools is needed.

Here, we present the results of a comparative multi-laboratory study that provides a systematic evaluation of eleven analysis tools (summarized in Table 1) using simulated as well as experimental data of varied complexity. Three of the analysis tools were utilized under different conditions, leading to a comparison of 14 different analyses. While clearly not all existing analysis tools could be covered (new tools are released continuously), this blind study (illustrated in Fig. 1) allows us to directly assess the performance of the different analysis approaches for the inference of kinetic information from single-molecule FRET trajectories and to identify their strengths and weaknesses. Specifically, we assess the accuracy of the inferred kinetic model (i.e., the kinetic rate constants and their connectivity) plus the associated uncertainties, and this for kinetic models of varied size, from the simple case of a two-state system (Fig. 2) to the more complex case of a non-equilibrium three-state system (Fig. 3), and finally to degenerate multi-state systems (Figs. 4, 5). All analyses were performed by the expert labs of each tool to ensure optimal implementation (see Methods for details).

## Results

### The archetypal 2-state system

We first consider the simplest case of a kinetic 2-state system, which could represent alternation between two conformations of a biomolecule in dynamic equilibrium, or transient biomolecular interactions. The kinetics of this system are described by two rate constants (Fig. 2a). In a blind study, we analysed simulated and experimental smFRET data using the diverse set of analysis tools summarized in Table 1 and detailed in the Supplementary Methods. Simulated test data (described in Methods) has the advantage that the underlying ground truth (GT, i.e., the simulation input) is known, which facilitates the evaluation of the inferred results, while, for experimental data, the GT is naturally not known. Figure 2b depicts an example of the simulated traces. We note that it closely resembles the experimental trace in Fig. 2e. Based on a dataset of such simulated traces ($n = 75$), all laboratories inferred FRET efficiencies (Fig. 2c) and rate constants (Fig. 2d), which agree very well: the FRET efficiencies deviate by less

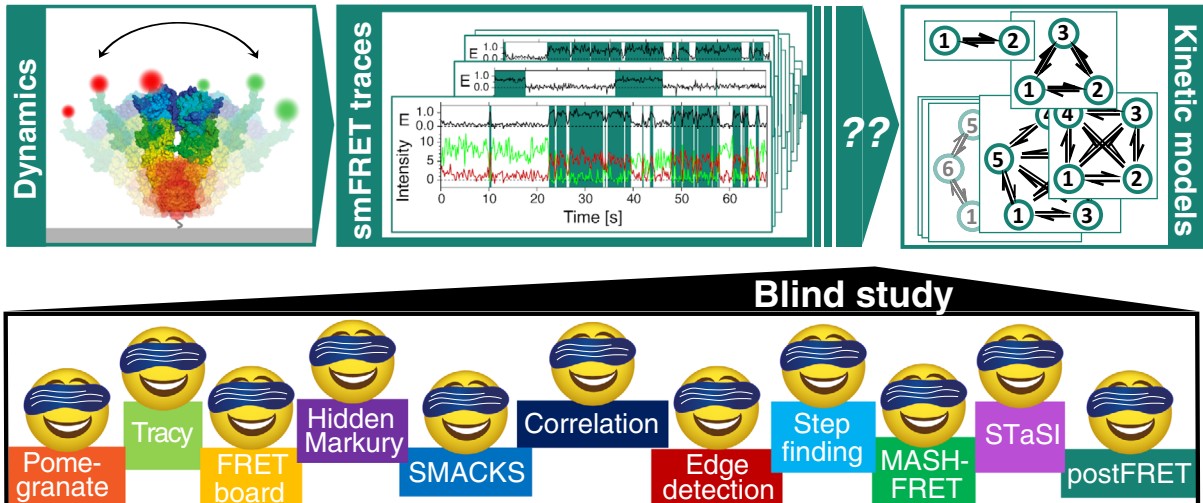

**Fig. 1 | This blind study reports on the performance of diverse analysis tools to describe single-molecule dynamics with quantitative kinetic rate constants.** Biomolecular dynamics of proteins and nucleic acids can be detected by smFRET and other single-molecule techniques. Extracting testable kinetic rate models from the experimental time traces is complicated by experimental shortcomings. Multiple labs joined forces to directly test the performance of diverse analytical approaches to infer kinetic rate constants in a blind study.

**Table 1 | Overview of the kinetic analysis approaches and software tools considered herein, grouped based on similarity**

| # | Tool name | Keywords | Description | Model selection | Uncertainty |
|---|---|---|---|---|---|
| 1 | Pomegranate | HMM | The python package Pomegranate is used for efficient and iterative modeling, fitting and evaluation of state numbers using the BIC. Dwell time analysis is subsequently performed after defining all transitions using a multivariate Gaussian fitting scheme and unbinned maximum likelihood fitting. | BIC | 95% CI |
| 2 | Tracy | HMM | Global HMM analysis was performed while setting the FRET efficiency and sigma as parameters to be learned. The state transitions and the state dwell times are selected by the user in a transition density plot and fitted with an exponential to obtain the rate constants. | Manual inspection | 95% CI |
| 3 | FRETboard33[35] | Semi-supervision, remotely served | A semi-supervised classification tool served remotely through a browser window. Users supervise the training of a classification model of choice, by manually correcting classification of example traces until the quality of automated classification is satisfactory. | Path probability | 95% CI |
| 4 | Hidden-Markury | 1D/2D- HMM | Hidden-Markury is a trace analysis software based on an interactive Jupyter notebook script, supporting global 1D FRET efficiency traces or 2D donor & acceptor photon streams, optionally treating degenerate states, forbidden transitions, fixed model parameters. | BIC | Sub-sampling |
| 5 | SMACKS[13] | 1-3D- HMM | Semi-ensemble HMM is used to extract one kinetic model from many smFRET fluorescence traces without prior discretization in two steps: (1) per trace HMM optimization (2) global per dataset optimization of the kinetic model, with pre-trained intensity parameters. | BIC | 95% CI |
| 6 | SMACKS (SS) | 1-3D- HMM | Test for user bias in semi-supervised inference: independent second analysis using SMACKS by S. Schmid. | BIC | 95% CI |
| 7 | Correlation | Discretized correlation | An unbiased, model-independent approach to obtain quantitative relaxation times from the negative amplitude of the cross-correlation function[26,27]. To enable a quantitative analysis of multi-state systems, a filtered correlation analysis[52] is performed based on the state sequence obtained with a step-finding algorithm[53]. | BIC | 95% CI |
| 8 | Edge finding (CK) | CK filter | The Chung Kennedy non-linear filter is applied to the time records of donor, acceptor and/or FRET efficiency to identify state transition points as sudden increases in the standard deviation of points in forward/back-ward predictor windows. Transition edges are confirmed by a two-sample student's t-test on the forward/ backward windows. | Manual inspection | Not assessed |
| 9 | Edge finding(k-means) | k-means clusters | All data points in either the donor and acceptor or the FRET efficiency time traces are assigned to distinct clusters. The mean value of each cluster is calculated and the points are reassigned to clusters to iteratively minimize the differences between the point values. Transition edges are identified as cluster assignment changes. | Manual inspection | Not assessed |
| 10 | Step finding | Line fitting | The entire dataset is iteratively fit with an increasing number of line segments. The addition of line segments is accepted if the overall fit quality is improved significantly. Rate constants are derived from dwell time analysis of line segments, which are assigned to a FRET state based on their mean FRET efficiencies. (SEM: standard error of the mean.) | Manual inspection | SEM / 68% CI |
| 11 | STaSI | Student's t-test | Detects step transition using the Student's t-test. Segments are grouped into states by hierarchical clus-tering. The optimum number of states is established using a minimum description length equation that sums the goodness of fit measured using the L1 norm to consider the sparseness of the states and transi-tions. (MDL: minimum description length.) | MDL | 95% CI |
| 12 | MASH-FRET (bootstrap)[54] | STaSI, vbFRET, bootstrap | A MATLAB-based GUI for the simulation and analysis of smFRET videos and fluorescence time traces[55]. Initial FRET states are obtained using STaSI and a BIC selection on 2D-Gaussian mixtures that model the global transition density plot. Refined FRET states, transition rate constants and uncertainties are then obtained using vbFRET and single exponential fit on bootstrapped dwell time histograms. | BIC | Standard deviation (2σ) |
| 13 | MASH-FRET (prob.)[56] | STaSI, vbFRET, DPH | The degeneracies of FRET states are estimated from ensemble dwell time histograms by performing a BIC selection on phase-type distributions. The fix-sized transition rate matrix is finally optimized using the Baum-Welch algorithm on hard-assigned FRET state trajectories. | BIC | 95% CI |
| 14 | postFRET | Monte Carlo | Simple thresholding is used for an initial assessment of the rate constants. A computationally-intensive Monte Carlo simulation is then used to find simulated trajectories that contain the same rate and error pattern as the experimental ones to guess a possible truth. Compare the two and adjust the guess for the next iteration. Noisy data is binned for the initial thresholding. (LAD: least absolute deviations). | LAD | 68% CI |

All tools are detailed in the Supplementary Methods.
BIC Bayesian information criterion, CI confidence interval.

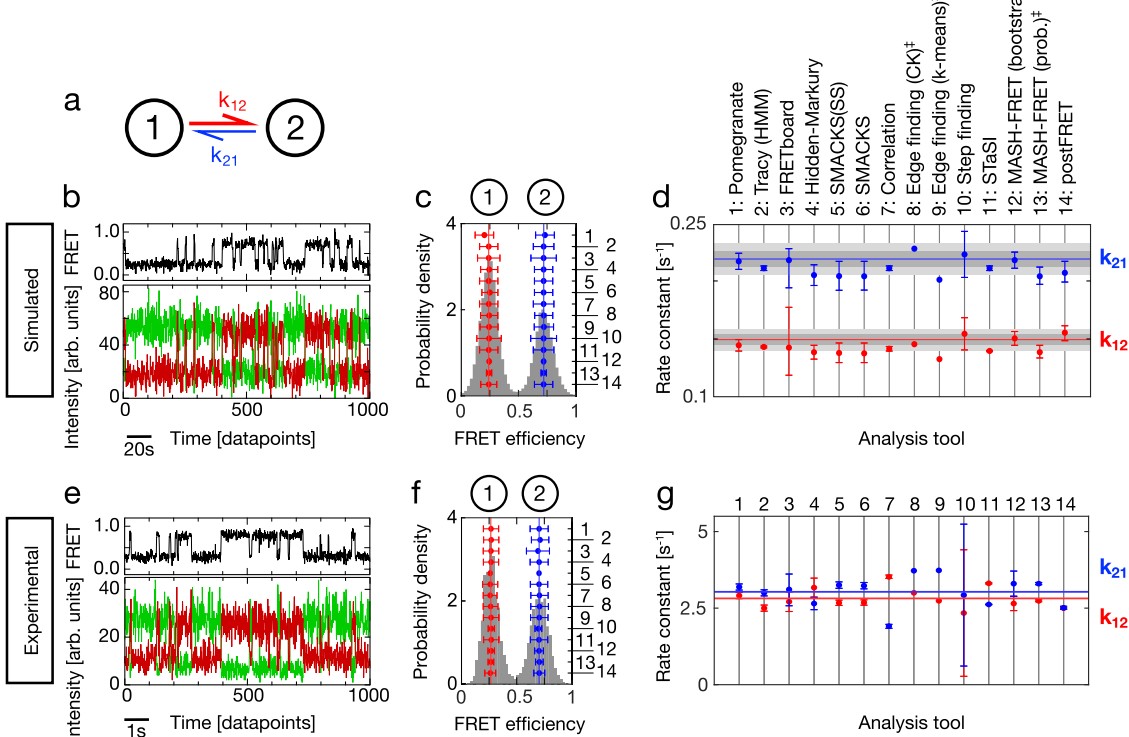

**Fig. 2 | Quantification of simulated and experimental kinetics between two states. a** Illustration of the kinetic model with two states (circles) connected by forward and backward rate constants: $k_{12}$ and $k_{21}$. **b** A simulated FRET trace showing the donor and acceptor fluorescence intensity (green, red) and the FRET efficiency (FRET, black), representative for the dataset used in (**c, d**): n(traces) = 75, n(datapoints) = 59,486, sampling rate = 5 Hz, time per datapoint = 200 ms. **c** FRET efficiency histogram (gray) with assigned states on top and inferred FRET efficiencies in red and blue. Numbers on the right axis refer to the analysis tools specified in (**d**). Vertical lines indicate the mean over all tools. Sample size as in (**b**). The error bars represent standard deviations. **d** Rate constants and uncertainties inferred from the dataset in c by different labs using the respective analysis tools. The ground truth (GT) is indicated by horizontal red and blue lines, the intrinsic uncertainty of the dataset (see text) is represented by dark gray (1σ) and light gray (2σ) intervals. Sample size as in (**b**). Uncertainty measures (CI, SD) as listed in Table 1. **e** An

experimental time trace with colors as in (**b**), representative for the dataset used in (**f, g**) with n(traces) = 19, n(datapoints) = 226,100, using 10 ms time bins resulting in 100 Hz sampling, kindly provided by B. Schuler. **f** FRET histogram with color code and axis labels as in (**c**). Sample size as in (**e**). The error bars represent standard deviations. No uncertainties were submitted for tool #5. **g** Inferred rate constants from the experimental dataset in (**f**). Color code as in (**d**). Horizontal red and blue lines indicate the mean of the inferred rate constants. Sample size as in e. Uncertainty measures (CI, SD) as listed in Table 1. Supplementary Fig. 2 shows the experimental data and analysis with ten times higher time resolution. ‡ denotes results that were submitted after the GT was known. The model size was restricted to two states. *FRETboard* and *Step finding* found erroneously large uncertainty intervals, which has been corrected in their latest software versions. See Supplementary Datafiles. Source data are provided as a Source Data file for panels (**c, d, f,** and **g**).

than 17% from the GT (1% average deviation), and the inferred rate constants deviate with a maximum of 12% from the GT (5% average deviation), with a slight systematic underestimation in most cases, i.e., the determined rate constants were slower. *Pomegranate*, *FRETboard*, and *Step finding* infer the most accurate rate constants under the tested conditions (Table 3). The equilibrium constants $K = k_{21}/k_{12}$ vary generally less since systematic deviations balance each other in this case (Supplementary Fig. 1a). In contrast, the reported uncertainty measures vary greatly, independent of the analysis type (0.4% to 21% relative to the inferred rate constant). For comparison, we estimated the minimal uncertainty given the finite size of the dataset, by quantifying the standard deviation of the rate constants obtained from one million simulated samples (see Methods). This standard deviation is ≥3% of the rate constants for the provided dataset (gray and light gray bars in Fig. 2b shown for 1σ and 2σ, respectively). Thus, most analysis tools reported reasonable uncertainty estimates, while some tools reported uncertainties that are smaller than this lower limit (*Tracy*, *Correlation*, *STaSI*) or provided no uncertainty measures (*Edge finding*). *FRETboard* version 0.0.2 reported consistently very large uncertainties, which was solved in their latest software version 0.0.3 (ref. 35, cf. Supplementary Datafiles). *Step finding* version 0.0.1 initially found erroneously large uncertainties that have been corrected in the latest software version 0.0.2 (cf. Supplementary Datafiles). We note that

various methods are currently in use for estimating uncertainties which complicates the direct comparison.

Next, we consider experimental data (see Methods), which naturally contains all typical noise sources and experimental artefacts (Fig. 2e–g). As there is no GT for experimental data, we assessed the consistency of the inferred FRET efficiencies and rate constants using the coefficient of variation (CV, i.e., the standard deviation divided by the mean). We found excellent agreement for all inferred FRET efficiencies (CV ≤ 2%). The rate constants vary by 12% and 16% (CV for $k_{12}$ and $k_{21}$, respectively), consistent with the variation found for simulated data (Fig. 2d). Again, no correlation of the rate constants with respect to the analysis approach is evident, but the tendency of a given tool for large or small uncertainties is conserved (Fig. 2d, g), with *FRETboard* and *Step finding* reporting the largest uncertainties, and *STaSI*, *MASH-FRET* (prob.), *postFRET*, and *Correlation* the smallest uncertainties. In most cases, the equilibrium constants (Supplementary Fig. 1b) agree well with each other and with the equilibrium populations of the FRET histogram, while some results are inconsistent with the latter (*Hidden-Markury*, *Correlation*, *STaSI*, and *postFRET*).

One important factor in dynamic smFRET data is the signal-to-noise ratio (SNR), which depends on the acquired signal per data point and can be controlled by the integration time (also known as exposure time). We explicitly tested the effect of a ten-fold shorter integration

time. On the one hand, this offers better sampling of fast kinetics due to the increased time resolution (1 kHz instead of 0.1 kHz sampling), but, on the other hand, it results in a lower signal-to-noise ratio which is more challenging for state identification. In addition, at 1 kHz sampling, the data shows single-photon discretization and non-Gaussian noise (Supplementary Fig. 2a, b), thus deviating from the basic assumptions underlying most of the considered analysis tools. Indeed, the overall agreement of the rate constants at this lower SNR was reduced: CV = 33% and 45% for $k_{12}$ and $k_{21}$, respectively (Supplementary Fig. 2c), indicating that the benefit of the increased time resolution is minor in this case. Nevertheless, the equilibrium constants agree very well again (CV = 2%, when excluding the two clear outliers in Supplementary Fig. 2d) due to the cancelation of systematic shifts for both rate constants (Supplementary Fig. 2e). Comparing the rate constants inferred at 1 kHz and 0.1 kHz sampling, *pomegranate*, *Tracy*, *Correlation*, *MASH-FRET*, and *Step finding* reported similar values (Supplementary Fig. 2e), while *STaSI* inferred slower rate constants for faster sampling. Conversely, *FRETboard*, and *SMACKS* inferred faster rate constants for faster sampling, either due to fitting noise or due to short events that are missed at lower time resolution. The latter is less plausible, given that the inferred rate constants are 20-fold smaller than the 0.1 kHz sampling rate. Thus, a comparison between 0.1 kHz and 1 kHz sampling can serve to estimate the robustness of the analysis tools towards non-Gaussian noise. Taken together, fundamentally different analysis approaches inferred consistent rate constants and FRET efficiencies from a simple, two-state system both for simulated data and experimental data with varied SNR.

## Directional sequences in a non-equilibrium steady-state system

Many biomolecular systems involve more than just two functionally relevant states, leading to more intricate kinetic models with more rate constants and, hence, more degrees of freedom. Such systems with three or more states can show a conceptually unique thermodynamic phenomenon: the non-equilibrium steady-state, in which a biomolecule, such as a motor protein or a molecular machine such as $F_0F_1$-ATP synthase, is driven by continuous external energy input, e.g. in the form of a chemical gradient[36], light[37,38], or ATP. As a result, conformational states may appear in a preferred sequence order, causing a non-zero net flow, e.g. for the 3-state system depicted in Fig. 3a:

$$\Delta G_{1\to 2\to 3\to 1} = -k_B T * \ln\left(\frac{k_{21} \cdot k_{32} \cdot k_{13}}{k_{12} \cdot k_{23} \cdot k_{31}}\right) \neq 0 \qquad (1)$$

The unique ability to directly observe the non-equilibrium steady-state is a prime example of the merits of single-molecule studies. Hence, we investigated it explicitly, using smFRET data simulated with a kinetic 3-state model and a non-zero counter-clockwise flow: $\Delta G_{1\to 2\to 3\to 1} < 0$ (Fig. 3a, b). As an additional challenge, this data contained fluorescence intensity variation between individual dye molecules, as observed in experimental data due to varied local dye environment and orientation, inhomogeneities in excitation intensity and polarisation, and also variations in detection efficiency[39].

All analysis tools found the three clearly separated FRET efficiency populations (Fig. 3c), while the inferred rate constants varied more than for the 2-state systems above (Fig. 3d). Most tools systematically underestimated $k_{13}$ and $k_{31}$ and overestimated all other rate constants. This may be attributed to the inevitable effect of time discretization and related intensity averaging: time-weighted averaging (e.g. camera blurring) of the FRET efficiencies can lead to mid-FRET observations that are indistinguishable from those caused by a bona fide biomolecular conformation. While, at the single datapoint level this discretization artefact cannot be prevented, the inference accuracy may

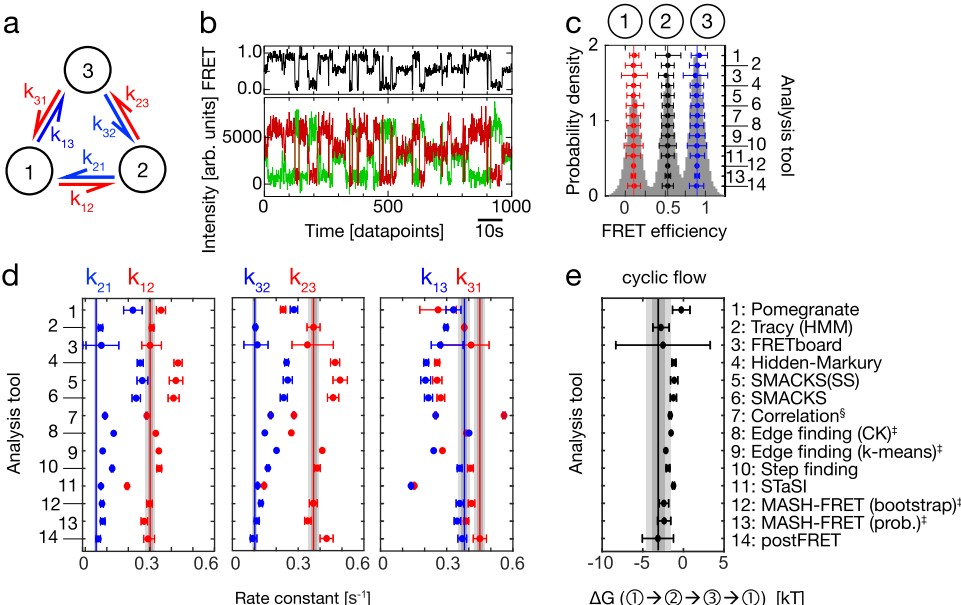

**Fig. 3 | Quantitative analysis of a non-equilibrium steady-state system. a** An illustration of the simulated three-state model with a counter-clockwise net flow. States (circles) are connected by forward and reverse rate constants as specified. **b** A simulated smFRET trace with donor and acceptor fluorescence intensity (green, red) and FRET efficiency (FRET, black), representative for the dataset used in (**c**, **d**, **e**): n(traces) = 150, n(datapoints) = 82,594, sampling rate = 10 Hz, time per datapoint = 100 ms. **c** SmFRET histogram overlaid with the inferred FRET efficiencies (right axis, numbers as in **e**) and assigned states on top. Sample size as in (**b**). The error bars represent standard deviations. **d** Inferred rate constants are shown in red and blue as specified. Vertical lines indicate the GT. The intrinsic uncertainty of the dataset is represented by dark gray (1σ) and light gray (2σ)

intervals. Sample size as in (**b**). Uncertainty measures (CI, SD) as listed in Table 1. Analysis tools are numbered as in (**e**). **e** The inferred cyclic flow in the counter-clockwise direction determined by calculating ΔG from Eq. (1) and compared with the GT value (solid vertical line). The uncertainty intervals (dark and light gray) are plotted as in (**d**). Sample size as in (**b**). Uncertainty measures (CI, SD) as listed in Table 1. Additional simulations to validate the dataset are shown in Supplementary Fig. 3. ‡ denotes results that were submitted after the GT was known. Edge finding did not report uncertainties. § denotes that the misassignment of start and end states was corrected after the GT was known. Source data are provided as a Source Data file for panels (**c**, **d**, and **e**).

be improved by treating discretization-induced averaging explicitly in the analysis[33,40]; or using pulsed illumination to reduce blurring[41,42]. Overall, *postFRET* and *Tracy* inferred the most accurate rate constants with average GT deviations of 9% and 14%, respectively. As shown in Fig. 3e, qualitatively, the net flow was correctly resolved (most accurately by *postFRET*, *Tracy*, and *FRETboard*), while quantitatively it was mostly underestimated, which we attribute to the aforementioned systematic misallocation of transitions between states 1 and 3. For this simulated dataset, the theoretical lower limit of the uncertainty (as introduced above for the 2-state system) is smaller because the dataset is larger. About half of the tools reported uncertainties that are in line with this lower limit (grey intervals), while the other half reported none or too small uncertainties. We would like to stress that such a quantification of net flow is only meaningful when no detailed balance constraints are imposed during the rate inference, which was the case for the tools considered here. Altogether, the rate constants of the non-equilibrium 3-state system with intensity variation were less accurate than those of the 2-state system, and also the uncertainty estimation was challenging in this case. Nevertheless, the steady-state flow was qualitatively well resolved by most tools.

## States with overlapping FRET efficiencies

Many biological systems show multi-exponential dwell-time distributions with long and short dwell times for the same apparent FRET state[6,43–45]. This can, for example, arise when the one-dimensional reaction coordinate spanned by the FRET pair is not sufficient to uniquely identify structural states in 3D space. Such kinetic heterogeneity is difficult to interpret because transitions between states with identical or overlapping FRET efficiencies cannot be directly observed in the recorded time traces, while they can often be inferred kinetically. To investigate this case, we simulated kinetic heterogeneity based on a four-state model (Fig. 4a) where states 1 and 2 have the same low-FRET efficiencies, and states 3 and 4 have the same high-FRET efficiencies. Again, the fluorescence traces included intensity variations between FRET pairs as observed in the experiment (introduced in the previous section), and also donor and acceptor blinking was included, as an additional imperfection of the data. Figure 4b shows example traces from the simulation and Fig. 4c shows the FRET efficiency histogram with two peaks. Without a priori knowledge of the model size, most tools identified the correct number of two apparent FRET states, while *FRETboard* used three FRET states to describe the data. *Edge finding* was not developed to deal with such kinetic heterogeneity, and *Pomegranate, Correlation, STaSI* and *MASH-FRET (bootstrap)* reported FRET efficiencies but no kinetic models. In the following, we use cumulative dwell-time distributions derived from each inferred model (Fig. 4d, detailed in Methods) to compare models with the correct number of FRET states but differences in the kinetic model, such as the connectivity of states or the number of hidden states (rate constants of all inferred models are reported in the Supplementary Table 1, and in the Supplementary Datafiles). Out of the seven independently inferred

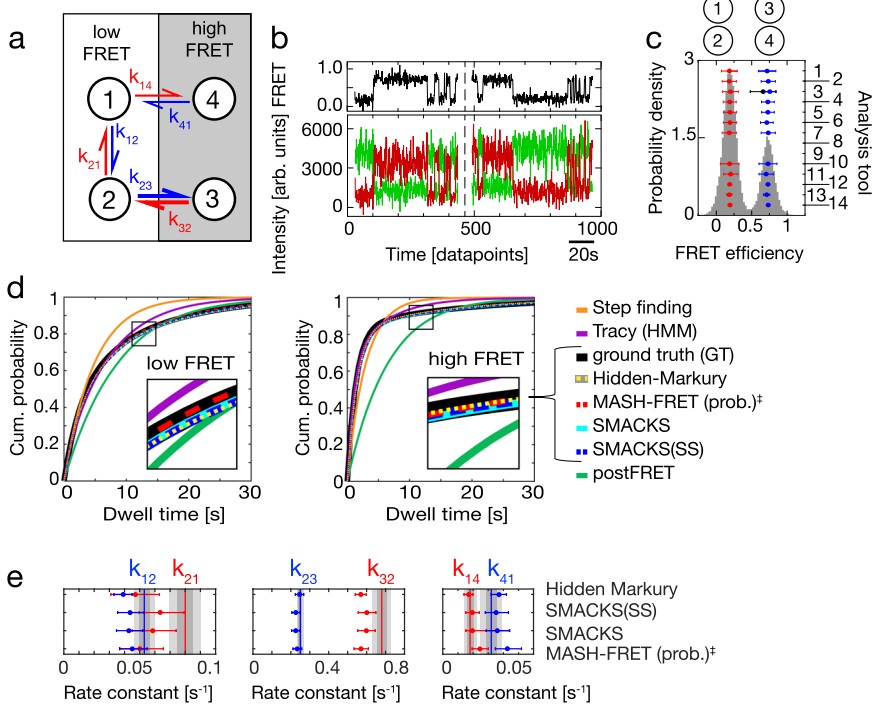

**Fig. 4 | Resolving kinetic heterogeneity: states with indistinguishable FRET efficiencies but different kinetics. a** An illustration of the simulated GT model with states (circles) connected by forward and reverse rate constants. States 1 and 4 as well as states 2 and 3 have indistinguishable FRET efficiencies, causing kinetic heterogeneity. **b** Two simulated FRET traces offset in time with donor and acceptor fluorescence intensity (green, red) and FRET efficiency (FRET, black) are shown, representative for the dataset used in (**c, d**): n(traces) = 250, n(datapoints) = 56,794, sampling rate = 5 Hz, time per datapoint = 200 ms. **c** FRET histogram with inferred FRET efficiencies overlaid (right axis: legend as in Table 1 and in all Figures). Sample size as in (**b**). The error bars represent standard deviations. **d** Comparison of cumulative dwell time distributions derived from the kinetic models with two FRET states (detailed in Methods). The GT histogram is shown as a bold black line. Insets show zoomed-in views of the data indicated by the squares. **e** Quantitative

comparison of the four most accurately inferred kinetic models: the GT values are represented as red and blue vertical lines. Sample size as in (**b**). Uncertainty measures (CI, SD) as listed in Table 1. The intrinsic uncertainty of the dataset is shown as dark gray (1σ) and light gray (2σ) intervals. Beyond the six displayed rate constants, these additional rate constants were inferred: for Hidden Markury $k_{31} = 0.045$ and $k_{34} = 0.003$, for SMACKS $k_{13} = 0.0001$, $k_{31} = 0.0055$, $k_{34} = 0.0034$, for MASH-FRET (prob.) $k_{31} = 0.033$. All inferred values of all models are reported in the Supplementary Tables 1 and in the Supplementary Datafiles. ‡ denotes results that were submitted after the GT was known. No results were reported by *Edge finding*. Participants were informed that kinetic heterogeneity may be involved, but not in which configuration. Source data are provided as a Source Data file for panels (**c, d,** and **e**).

kinetic models, the two models without kinetic heterogeneity (by *Step finding* and *postFRET*) show the largest deviations from the GT, as these models cannot reproduce the multi-exponential nature of the dwell-time distribution. On the other hand, the four models inferred by the HMM-based *Hidden-Markury, SMACKS, SMACKS(SS)*, as well as *MASH-FRET (prob.)* show good agreement with the GT and overlay the GT in the low- and high-FRET case (compare Fig. 4d). A quantitative comparison of these four models and their uncertainties with the GT is provided in Fig. 4e. It shows accurate rates and some collective underestimation of rates $k_{12}$, $k_{21}$, $k_{32}$, likely due to missed fast events. In addition, some rates were inferred that are not present in the GT (see Fig. 4e caption). Taken together, several tools inferred the correct model size (number of states) and accurate cumulative dwell-time distributions, but model selection – and in particular the selection of the correct connectivity of states – remains a main challenge in inferring kinetic information from smFRET trajectories. It is, however, encouraging that several analysis tools can already deduce kinetic models that closely reproduce the GT even under difficult conditions involving kinetic heterogeneity.

## Full complexity of a black-box experiment

Encouraged by the previous results, we tested all tools vis-à-vis the full experimental complexity to see if they perform similarly as in the simulated case (Fig. 4). Three experimental datasets of the same

biological system (protein binding to a fluorescently labelled DNA, see Methods), under different experimental conditions and thus different kinetic behaviour, served as a test case. However, the analysts had no prior information on the molecular system causing the dynamics. This means that all the effects discussed so far could potentially be present in these experimental datasets: multiple FRET states, diverse noise sources, fluorophore blinking, directional steady-state flow and kinetic heterogeneity. In addition, the fluorescence intensity variation between single molecules was particularly high in these datasets (see Fig. 5a, d, g), which complicated the inference of the number of states and rates involved (subsequently referred to as model selection). Under these complex conditions, the inferred number of FRET states (Fig. 5b, e, h) varied more than in the simulated case (discussed in Fig. 4). Most tools found two FRET states (Fig. 5b, e, h, some of them including kinetic heterogeneity), but also three, four, or more different FRET states were reported (Supplementary Fig. 4), and the kinetic rate constants varied accordingly. Given the inherent lack of GT information in experimental data, we cannot quantitatively assess the accuracy in this comparison. To balance this fact, we qualitatively compare the inferred results for all three datasets. The 6–7 models with two FRET states (and possibly more hidden states) are compared in (Fig. 5c, f, i). Other models with three, four, or more FRET states are compared in Supplementary Figs. 4–6. (All inferred rate constants are given in Supplementary Tables 2–4 and Supplementary Datafiles). Again, we

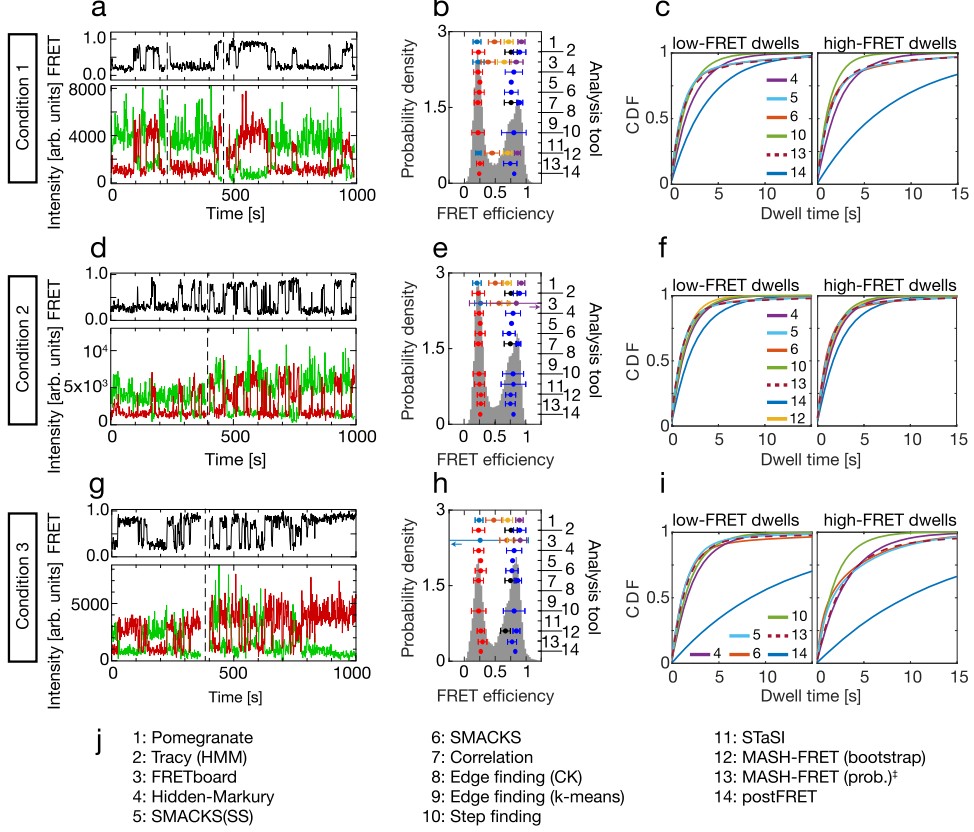

**Fig. 5 | Increased experimental complexity.** Results inferred from three experimental datasets where naturally no GT exists. **a, d, g** Experimental traces, offset in time and separated by dashed vertical lines, with donor and acceptor fluorescence intensity (green, red) and FRET efficiency (FRET, black), representative for the datasets used in (**b, c**), (**e, f**), (**h, i**), respectively, with n(traces): 134, 163, 118; and n(datapoints): 36,604, 37,067, 43,512; sampling rate = 33 Hz, time per datapoint = 30 ms. All three datasets were kindly provided by M. Schlierf. **b, e, h** FRET efficiency histograms and FRET efficiencies inferred by the analysis tools numbered as in (**j**). Sample sizes as in (**a, d, g**), respectively. The error bars represent standard deviations. For clarity, only the smallest reported model is shown for each analysis tool,

up to a maximum of four FRET states. All inferred FRET efficiencies are shown in Supplementary Fig. 4, and all inferred results are provided in the Supplementary Tables 2–4 and in the Supplementary Datafiles. Purple arrow in (**e**): the error bar extends to 1.61. Teal arrow in (**h**): the error bar extends to −0.53. **c, f, i** Cumulative distribution functions (CDF) of the dwell-times simulated using the inferred kinetic models with two FRET states, obtained with the tools numbered as in (**j**). **j** Legend with all analysis tools. No results were reported by *Edge finding*. ‡ denotes results that were submitted after all other results were known. Source data are provided as a Source Data file for panels (**b, c, e, f, h,** and **i**).

use cumulative dwell-time distributions (cf. last section) derived from each inferred model (Fig. 5c, f, i) to facilitate the comparison of models with the same number of FRET states but possibly different state connectivity. The distributions are thus single- or double-exponential depending on the reported kinetic model. The five tools that inferred two FRET states and qualitatively similar kinetic models under all three conditions despite different analysis approaches, are the HMM-based *Hidden-Markury* and *SMACKS*, as well as *Step finding, postFRET and MASH-FRET (prob.)*. While *postFRET* consistently inferred slower rate constants, the qualitative agreement among the other five tools is surprisingly good (CV ≤ 25% for the average residence time) despite the complexity of the input data, the missing prior knowledge about the system, and the different analysis approaches used.

Altogether, we conclude that model selection and state allocation are currently the key challenges in the analysis of kinetic data. In this study, we focused only on the analysis of fluorescence intensity and FRET efficiency data. The addition of complementary information from simulations or experiments (e.g., static molecular structures and other observables, such as fluorescence lifetimes, anisotropy, and more) may help to elucidate complicated or otherwise under-determined systems[30,46,47].

## Discussion

In this blind study, we compared eleven kinetic analysis tools for the inference of quantitative kinetic rate constants based on single-molecule FRET trajectories. We explicitly considered the major (kinetic) challenges that the single-molecule experimentalists are typically confronted with: determining the best model to describe the data, especially with multiple FRET states, a varying signal-to-noise ratio, directional non-equilibrium steady-state flow, and kinetic heterogeneity (i.e., states with indistinguishable FRET efficiency but distinct kinetics). We assessed the inferred FRET efficiencies, rate constants, and the reported uncertainties, based on three simulated datasets and four experimental datasets from two biological systems measured using two different setups in different laboratories. The simulated data allowed us to directly assess the accuracy of the inferred rate constants using the known ground truth model and to judge the plausibility of the reported uncertainty measures, while the experimental data shows the relevance and validity of this study.

We found that the number of states was correctly inferred by all tools, as long as their FRET efficiencies were clearly separated (Figs. 2 and 3). In the presence of kinetic heterogeneity with overlapping FRET states, model selection was more challenging (Fig. 4). In this case, three tools successfully inferred models that accurately reproduce the dwell-

time distribution of the GT despite overlapping FRET states (*Hidden-Markury, MASH-FRET, SMACKS*). In general, the accuracy of the rate constants inferred by all tools decreased with increasing model size and complexity, where time discretization artefacts and inter-trace intensity variation become increasingly challenging. The equilibrium constants and steady-state flow were more accurately inferred than individual rate constants due to the cancellation of systematic errors (Supplementary Figs. 1 and 2d, e, Fig. 3). Caution is advised with the uncertainties of rate constants since different uncertainty measures are reported by different approaches. Even for small models (Figs. 2 and 3), we found that some uncertainty estimates were smaller than the uncertainties caused by the finite dataset size, while interestingly, more plausible uncertainties were reported for the more complex model in Fig. 4 (Supplementary Fig. 4). In general, the comparison of uncertainties is complicated by the fact that no common standard exists and the mathematical interpretation of the reported uncertainty intervals differs from tool to tool.

When comparing various analysis frameworks, model-free approaches are generally considered advantageous for an unbiased data analysis. However, HMM-based tools (that compare several input models based on scarcity criteria) were found to be more robust towards data heterogeneity (Figs. 4 and 5, Supplementary Fig. 2). Nevertheless, we did not observe a clear overall clustering of the inferred rate constants with the underlying analysis framework, likely due to differences in the data handling beyond the used algorithms (e.g. supervised, semi-supervised, or unsupervised inference). The total analysis durations (processing and computation) ranged from a few minutes to several hours depending on the analysis tool and the model size, with *StaSI* and *Step finding* ranking among the fastest, and *SMACKS* among the slower tools. In the course of this study, multiple conceptual oversights could be found and solved in a number of tools, which is a direct constructive result of this collaborative comparison study that led to the general recommendations stated in Table 2. Additionally, a simple shareable smFRET data format was introduced (Supplementary Note 1) and utilized by all twelve labs working in diverse software environments. We anticipate that this data format will facilitate future collaborations and significantly lower the barrier for an experimentalist to adopt a newly developed analysis tool if it supports the accepted format.

Looking ahead, a particularly promising outlook is the possibility to characterize individual states with individual noise patterns more accurately, using machine learning. Recently, deep learning approaches have been developed for the unbiased selection of single molecule traces for further kinetic analysis[48,49]. Similar approaches could be

## Table 2 | General recommendations for users and developers of kinetic inference tools[a]

| | |
|---|---|
| (i) | As a general consistency test, the inferred kinetic model (connectivity and rate constants) can be simulated and the output of the simulation compared to the original input data. For example, the simulator used herein is publicly available as a simple and powerful (MATLAB) tool to test whether the proposed model can generate data analogous to the original input, e.g. regarding FRET histogram, smFRET traces, etc. |
| (ii) | Potential biases in the analysis (e.g. regarding model size, state occupation, etc.) can be revealed by subjecting the re-simulated data (with known ground truth) to the same analysis approach as the experimental data. |
| (iii) | Where possible, kinetic models with a specific number and connectivity of states are preferred over mean residence times, since the latter leave the individual transition rate constants undetermined for more than 2 states. |
| (iv) | Uncertainty measures are necessary indicators of significance, and a unified standard would greatly improve their comparability. The 95% confidence interval was the most frequently used uncertainty measure in this study, and we encourage its use as a common standard for the future. |
| (v) | Benchmarking new analysis tools using datasets of varied complexity – including models with more than 2 states – can reveal systematic errors, e.g. regarding the weighting of multiple rate constants that depopulate a given state, an issue encountered in this study. |
| (vi) | Benchmarking new software with established test data helps the potential users to judge the added benefits of newly introduced analysis tools. The diverse datasets used herein are publicly available and can serve to assess a tool's performance under varied experimental conditions. |
| (vii) | Supporting broadly accepted file formats for newly developed analysis tools facilitates fast dissemination in the field. We offer the simple format described in Supplementary Note 1, which proved to be very useful for this study. |

[a]In the course of this study, several difficulties with the analysis of kinetic data have become apparent. Out of this experience, we have compiled a list of recommendations for those developing and using kinetic analysis tool.

**Table 3 | Summary of the test conditions for the individual datasets, including the prior information on ground truth (GT) and number (N) states, as well as three data characteristics: kinetic heterogeneity, photo-physics, and signal-to-noise ratio (SNR)**

| | GT known?[a] | N states predefined? | Kinetic heterogeneity | Photo- physics[b] | SNR[c] |
|---|---|---|---|---|---|
| Fig. 2 (sim.) | No | Yes, 2. | No | Clean | 4 |
| Fig. 2 (exp.) | No | Yes, 2. | Not observed | Mainly clean | 4 |
| Fig. 3 (sim.) | No | No | No | Intensity variation | 3 |
| Fig. 4 (sim.) | No | No | Yes | Intensity variation & blinking | 4 |
| Fig. 5 (exp.) | No | No | Yes observed | Intensity variation & blinking | 3 |

[a]Exceptions are labelled with a dagger in all figures.
[b]See simulation parameters in Supplementary Table 5.
[c]The SNR was obtained from the FRET efficiency histogram using Gaussian fits and SNR = $|\mu_1 - \mu_2|/\sqrt{\sigma_1^2 + \sigma_2^2}$.

envisioned for a model-free kinetic analysis, which bears the potential to improve model selection significantly[18,50]. Demonstrating such new tools using public training datasets and supporting the simple file format introduced in this study, will accelerate the dissemination of the newest theoretical developments within the community of single-molecule experimentalists. Additional important aspects for future benchmark studies include the 'data greediness' of a given tool, e.g. the amount of data and the number of transitions-per-trace (given by the relation of biomolecular dynamics versus bleach rate) that are required for accurate rate inference.

In conclusion, this blind study on kinetic inference from smFRET data further validates the use of smFRET in deciphering biomolecular rates. It unequivocally reveals the current strengths and weaknesses of the various analysis approaches when tested against frequently encountered phenomena in smFRET experiments, and provides a reference standard for the continuous developments in this active field. We anticipate that this study will serve the community as a guide for data interpretation, spark future developments in kinetic inference, and therefore help to advance our understanding of biomolecular dynamics leading to function.

## Methods

### Procedure of this benchmark study

The need for a comparison of analysis tools for smFRET trajectories has grown with the increasing number of smFRET users and published tools. This was addressed at the Biophysical Society Meeting 2019 (Baltimore, US) by initiating a kinetic software challenge, short kinSoftChallenge. In line with more efforts to assess, promote, and potentially standardize experimental and analytical smFRET procedures (Refs. 2,3, 51 in preparation), the kinSoftChallenge represents an important step aimed to improve the reliability and accuracy of kinetic inference from smFRET trajectories. In a first round of the study (July 2019 to November 2019), the participants received three simulated datasets (shown in Figs. 2, 3, and 4). In the second round (December 2019 to February 2020), the participants analyzed the experimental dataset shown in Fig. 5. Experimental data with high and low SNR was compared in a third round (November to December 2020, shown in Fig. 2, and Supplementary Fig. 2). The individual test conditions are described in the text and summarized in Table 3. All challenge rounds were conducted as blind studies, i.e., the participants did not have ground truth information during data analysis (exceptions are labeled with a dagger in all Figures).

### Simulation of smFRET trajectories

In short, simulated smFRET datasets were generated to mimic fluorescence traces obtained by TIRF-based experiments. State trajectories were modeled with a continuous-time approach and later discretized. Similar to experiments, this allows state transitions to occur during the integration time window (time bin of the detector). Noise was added to the fluorescence intensity traces using experiment-derived parameters to generate realistic data.

In more detail, for each molecule a continuous-time state trajectory was simulated based on the kinetic model, as specified by a transition rate matrix. A summary of the specific simulation parameters is given in the Supplementary Table 5 and all configuration files with all parameters are provided as Supplementary Datafiles. First, the trace length was determined from an exponential distribution described by the rate of photobleaching. The trace length was rejected if it was shorter than a minimal trace length and truncated to a maximal trace length (see Supplementary Table 5). Then, a random initial state was chosen based on the probability of being in a particular state given the transition rate matrix. Starting from this state, dwell times for all possible transitions to the other states were drawn randomly from exponential distributions defined by the transition rates, and the shortest dwell time determined the transition and the new state of the system. This process was repeated until the full trace length was reached. This state trajectory was then converted into discrete-time fluorescence intensity traces using a specified sampling rate. For each time bin (i.e., camera frame), the donor and acceptor intensities upon donor excitation and the intensity of the acceptor upon acceptor excitation were drawn from state-specific Gaussian distributions (specified by the means $\mu_I$ and covariance matrices given in the configuration file). The intensity in each channel during a time bin is given by the weighted average of all states visited during this specific time bin.

Typically, single-molecule fluorescence traces show variations in the fluorescence level between individual molecules, due to, amongst others, local variations in excitation power and local dye environment[13]. To take these variations into account, two additional sources of per-trace intensity variations were considered for the simulated data shown in Figs. 3 and 4. First, for each molecule, individual intensity levels for each state were chosen. To do so, the intensity level was drawn from an empirically determined state-specific Gaussian distribution (with mean $\mu_I$ and standard deviation $5*\sqrt{\mu_I}$). Second, for each molecule, an individual brightness factor was determined by $1.20^r$ where $r$ was randomly chosen from the interval $[-1, 1]$. Thus, this factor is distributed in the interval $[0.83, 1.20]$ and all channels were multiplied by the same factor. For the simulated data shown in Fig. 4, independent blinking of the donor and acceptor dye was modeled by a simple 2-state system ("bright", "dark"). In the case of an acceptor dark state, the FRET efficiency was set to zero. Details are given in Supplementary Table 5.

Five hundred additional datasets from the same parameter set were created and compared, to validate that the dwell time distribution of the dataset used in this study shows the expected behaviour (see Supplementary Fig. 3). Configuration files with all simulation parameters (including the ground truth for the kinetic models) for the synthetic data in Figs. 2, 3, and 4 can be found in the Supplementary Datafiles. The MATLAB scripts used for the simulation are publicly available at: www.kinSoftChallenge.com and https://doi.org/10.5281/zenodo.5701310. A Supplementary Table with the simulation

parameters and a Supplementary Note on the file format used herein are provided in the Supplementary Information file.

### Estimated minimal uncertainty of rate constants inferred from simulations

Because of the finite number of traces per datasets, only a limited random sample of dwell times is observed for each given transition, resulting in a variation of the rate constants inferred from different datasets with identical ground truth. In order to estimate this lower bound of the uncertainty for the inference of rate constants from a finite dataset, we randomly drew the same number of dwell times as provided in the simulated challenge dataset from an exponential distribution with time constant $\tau = 1/k$. The maximum likelihood estimator (MLE) for the rate constant that produced this set of dwell times $\Delta t$ is given by $1/\overline{\Delta t}$. This calculation of the MLE was repeated one million times. The standard deviation of these 1 million MLEs is a function of the number of dwell times present in the challenge data set – the more dwell times are observed, the narrower the MLE distribution – and hence, it depends on the transition rate constants and the total observation time. We used this standard deviation as an estimate of the lower bound for the uncertainty of inferred rate constants from the simulated datasets.

### Simulation of cumulative dwell-time distributions from inferred kinetic models

In order to compare submissions with the same number of FRET states but different underlying kinetic models (i.e., number of hidden states and connectivity), we simulated dwell times from the submitted kinetic models for the three datasets shown in Figs. 4 and 5. This yields cumulative dwell-time distributions that are characteristic for the kinetic model. Dwell times were accumulated from simulations of continuous time state trajectories (Supplementary Note 1) that included roughly 200x (Fig. 4d) or 400x (Fig. 5c, f, i) more time points than the original datasets.

### Origin of the experimental datasets

The experimental data shown in Fig. 2 and Supplementary Figs. 1, 2 was kindly provided by Benjamin Schuler. It shows the interaction between the nuclear-coactivator binding domain of CBP/p300 (NCBD) and the intrinsically disordered activation domain of the steroid receptor coactivator 3 (ACTR), measured using confocal single-photon detection[5]. The experimental data shown in Fig. 5 and Supplementary Fig. 4 was kindly provided by Michael Schlierf. It shows binding of single-strand binding proteins (SSB) to a fluorescently labelled DNA hairpin, measured in prism-type total-internal reflection fluorescence (TIRF) mode using camera-based detection (EMCCD)[4].

### Procedures of the kinetic analyses

Detailed descriptions of all analysis tools are provided in the Supplementary Methods in the Supplementary Information file. All inferred results are provided as Supplementary Datafiles.

### Reporting summary

Further information on research design is available in the Nature Research Reporting Summary linked to this article.

## Data availability

The simulated and experimental smFRET data used in this study are available at www.kinsoftchallenge.com and https://doi.org/10.5281/zenodo.5701310. All inferred results are provided in the Supplementary Data files. Supplementary figures, notes, and methods are provided in the Supplementary Information file. Source data are provided with this paper.

## Code availability

The simulation code and parameters to generate the simulated datasets are available at https://doi.org/10.5281/zenodo.5701310. All software tools are available: Pomegranate v0.0.1 at https://github.com/hatzakislab/DeepFRET-GUI; Tracy v4.4.8 upon request as it is being replaced by a new program for multi-color analysis (contact: Don C. Lamb [d.lamb@lmu.de], requests will be addressed as soon as possible, typically within 1 week); FRETboard v0.0.3 at https://github.com/cvdelannoy/FRETboard; Hidden-Markury v0.0.1 at https://github.com/ChristianGebhardt/Hidden-Markury; SMACKS v1.4 at https://github.com/sciSonja/SMACKS; Correlation v0.1b at https://doi.org/10.5281/zenodo.5512005; Edge finding (CK and k-means) v0.0.1 at https://www.physics.ncsu.edu/weninger/KinSoft.html; Step finding v0.0.2 at https://github.com/SMB-Lab/PyStepFinder; StaSI v0.0.1 at https://github.com/LandesLab/StaSI; MASH-FRET v.1.3.2 (bootstrap and probabilistic) at https://github.com/RNA-FRETools/MASH-FRET; post-FRET v4.0 at https://github.com/nkchenjx/postFRET.

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

## Acknowledgements

We thank Benjamin Schuler and Michael Schlierf for providing experimental smFRET data. We thank the 2019 chair of the Biological fluorescence subgroup of the Biophysical Society (BPS), Paul Wiseman, and the co-chairs of the FRET in Biophysics Discussion Forum, Claus Seidel and

Hugo Sanabria, for providing a platform to initiate the kinSoftChallenge. M.G. was funded by the Deutsche Forschungsgemeinschaft (German Research Foundation) – Project no. 431471305. R.K.O.S. thanks the SNF (200020_165868 and 200020_192153) and UZH for financial support. R.B. thanks the University of Applied Sciences Mittweida for financial support. L.V., J.S., T.H. were supported by the Collaborative Research Centre SFB1381 funded by the Deutsche Forschungsgemeinschaft (DFG, German Research Foundation) – Project-ID 403222702 – SFB 1381. S.S. was supported by the Postdoc.Mobility fellowship no. P400PB_180889 by the Swiss National Science Foundation. J.C. thanks US National Human Genome Research Institute (NHGRI) Project-ID 1R15HG009972. L.K. and D.D.M. thank the Case Western Reserve University College of Arts and Sciences for support. D.A.E. supported by NIH grant R35 GM127151. K.R.W. supported by NIH grants R01 GM132263 and R01 GM118508. C.D.L. and D.D.R. were supported by grant 16SMPS05 from the Institutes Organization of the Dutch Research Council (NWO-I, formerly FOM). N.S.H. acknowledges support from Carlsberg foundation Distinguished associate professor program (CF16-0797) Vellux foundation center of excellence BIONEC (grant no 18333) and the NovoNordisk foundation (NNF14CC00001 and NNF16OC0021948), G.H. and H.S. acknowledge support by NIH 1P20GM130451 and 2R01MHO 81923-11A1, and NSF 1749778. A.B. and C.A.M.S. acknowledge support by the European Research Council through the Advanced Grant 2014 hybridFRET (number 671208).

## Author contributions

All authors analysed data and/or discussed the results and contributed to writing the manuscript. M.G. and S.S. compiled all results, prepared figures, and wrote the article draft together with A.B. M.G. performed and analysed simulations. S.S. designed and initiated the study, and organized the collaboration in consultation with all authors.

## Competing interests

The authors declare no competing interests.

## Additional information

[1]Centre de Biologie Structurale, CNRS UMR 5048, INSERM U1054, Univ Montpellier, 60 rue de Navacelles, 34090 Montpellier, France. [2]Institut für Physikalische Chemie, Lehrstuhl für Molekulare Physikalische Chemie, Heinrich-Heine-Universität, Universitätsstr. 1, 40225 Düsseldorf, Germany. [3]Department of Chemistry & Nano-science Center, University of Copenhagen, 2100 Copenhagen, Denmark. [4]Novo Nordisk Foundation Centre for Protein Research, Faculty of Health and Medical Sciences, University of Copenhagen, 2100 Copenhagen, Denmark. [5]Department of Chemistry, University of Zurich, 8057 Zurich, Switzerland. [6]Department of Chemistry and Biochemistry, Ohio University, Athens, OH, USA. [7]Physical and Synthetic Biology, Faculty of Biology, Ludwig-Maximilians-Universität München, Großhadernerstr. 2-4, 82152 Planegg-Martinsried, Germany. [8]Department of Chemistry, University of North Carolina, Chapel Hill, NC 27599, USA. [9]Lineberger Comprehensive Cancer Center, University of North Carolina, Chapel Hill, NC 27599, USA. [10]Department of Physics and Astronomy, Clemson University, Clemson, SC 29634, USA. [11]Institute of Physical Chemistry, University of Freiburg, Freiburg, Germany. [12]Signalling Research Centers BIOSS and CIBSS, University of Freiburg, Freiburg, Germany. [13]Department of Physics, Case Western Reserve University, Cleveland, OH, USA. [14]Department of Chemistry, Case Western Reserve University, Cleveland, OH, USA. [15]Department of Chemistry and Center for Nano Science (CeNS), Ludwig Maximilians-Universität München, Butenandtstraße 5-13, 81377 München, Germany. [16]Bioinformatics Group, Wageningen University, Droevendaalsesteeg 1, 6708PB Wageningen, The Netherlands. [17]Department of Physics, North Carolina State University, Raleigh, NC 27695, USA. [18]Spemann Graduate School of Biology and Medicine (SGBM), University of Freiburg, Freiburg, Germany. [19]NanoDynamicsLab, Laboratory of Biophysics, Wageningen University, Stippeneng 4, 6708WE Wageningen, The Netherlands. [20]Present address: PicoQuant GmbH, Rudower Chaussee 29, 12489 Berlin, Germany. [21]Present address: Department of Bionanoscience, Kavli Institute of Nanoscience Delft, Delft University of Technology, Van der Maasweg 9, 2629 HZ Delft, The Netherlands. [22]Present address: Laserinstitut Hochschule Mittweida, University of Applied Sciences Mittweida, 09648 Mittweida, Germany. [23]Present address: Department of Biochemistry and Molecular Pharmacology, New York University School of Medicine, New York, NY 10016, USA. ✉e-mail: goetz@picoquant.com; schmid@nanodynlab.org

