## [Peer Review File · Nature Communications]

Reviewers' Comments:

Reviewer #1:

Remarks to the Author:

smFRET provides the unique possibility to obtain structural and functional data from single labeled biomolecules. However, the interpretation of the noisy fluorescence and FRET time traces in the framework of kinetic models is not simple and usually, biochemical and other spectroscopy data provide kinetic models which are used as basis for the interpretation of smFRET data. In order to obtain the number of states and the kinetic constants directly from smFRET data several groups have developed procedures to analyse such data sets. These groups initiated the „kinetic software challenge“ to evaluate and to standardize experimental and analytical smFRET procedures.

In this joint approach each group used its own analysis tool (11 different tools, three of them were used under different conditions). They analyse the same FRET data and determined the number of states and the kinetic constants. They use the following data sets for this comparison: a kinetic 2-state equilibrium state system with simulated test data and with an experimental data set; a kinetic 3-state non-equilibrium state with simulated test data; a kinetic 4-state model with overlapping FRET-efficiencies with simulated kinetic heterogeneity, and finally, three experimental data sets with overlapping FRET-efficiencies.

The results obtained with the different tools are summarized in four instructive figures showing the FRET efficiency histograms with the assigned states and the rate constants with their uncertainties. The details of these comparisons are given in the Supplementary Information. Based on these results the authors formulated six general recommendations for users and developers of kinetic inference tools including a simple file format for dissemination of smFRET data.

This study is the first comparison of different kinetic analysis tools for inference of kinetic rate constants from smFRET trajectories. It describes the strengths and weaknesses of eleven software tools and it provides a reference standard for future developments in this field. This joint approach summarizes the knowledge of the participating groups and will help newcomers in this field to select the appropriate tool for their data analysis.

The manuscript is clearly written, data files are provided in the supporting material, the results of the data analysis are presented in very instructive figures, data analysis and statistics are described in the different tools. This work provides reference standards for the analysis of smFRET trajectories and is benchmark for further developments.

I strongly recommend publication.

Minor remarks:

Unfortunately, there is no common standard for the uncertainties in the different tools what complicates the comparison of these results.

A description of each tool is given in the Supplementary Methods. These descriptions differ greatly in detail, some using half a page, some 5 pages. A consistent description would be very helpful as overview for non-experts in this field.

Reference 37 is wrong. It refers to the F1- ATPase where a three stepped rotation of an antibody bound to the gamma subunit driven by ATP hydrolysis was observed. The first observation of a three stepped rotation driven by a chemical gradient in ATP synthesis direction and in opposite direction during ATP hydrolysis was carried with the smFRET technique using a membrane integrated F0F1 ATP synthase. Diez et al. (2004) Nature Structural & Molecular Biology 11, 135-141

Reviewer #2:

Remarks to the Author:

This manuscript reports the results of a recent blind test study of the kinetic analysis of single molecule fluorescence data. Various simulated and experimental data were analyzed using 14 different methods by expert groups and the performances were compared. In addition to the analysis methods introduced and compared in this study, there must be other methods and variations, some of which are not publicly accessible. Various photon-by-photon analysis methods are not included here, either. However it would not be possible to cover everything, and I believe this is a timely and appropriate study that should be published. The blind data set was designed

well, including simulation (ground truth available) and real experimental data with a relatively easy, two-state system and more complex three- and four- state systems (both equilibrium and non-equilibrium systems). Overall, the analysis results are well summarized and presented. However, there are several models that cannot be analyzed well by some of the methods. I don't think this is not because those methods are poor or have problems, but because those methods are not compatible with the data model type. The authors could not know this before the manuscript was written. So I suggest adding more comments and explanations for the cases that the methods don't work well. See the comments below for more detailed suggestions and other minor issues.

1. Page 6, 2-state simulated data:

Usually, the errors of the rate coefficients are larger than those of the FRET efficiencies. However, the maximum deviation of the FRET efficiency 17% is larger than that of the rate 12% although the deviation of the FRET efficiencies seems very small in Fig. 2C and F. Therefore, I am not sure if comparing the maximum variation is right. What is the average deviation of the FRET efficiency? (This is 5% for the rate.) This could be a better comparison.

2. In Line 152. "FRETboard reported consistently very large uncertainties, which was solved in their latest software version 0.0.3 (Ref33). Step finding initially found erroneously large uncertainties that have been corrected in the latest software version."

In this case, it may be good to include the updated uncertainties by the latest versions in a note.

3. In Fig. 2, the dwell times and number of transitions within 100 s time window of the simulated data (Fig. 2b) and experimental data (Fig. 2e) are comparable. However, the extracted rate constants in Fig. 2d and g differ by more than a factor of 10. It looks like the y-axis in Fig. 2g was mislabeled.

4. In line 235, "Most tools systematically underestimated k_{13} and k_{31} and overestimated all other rate constants. This may be attributed to the inevitable effect of time discretization and related intensity averaging: when a transition between the high- and low-FRET states happens during a time bin, time-weighted averaging (camera blurring) of the FRET efficiencies occurs, leading, in some cases, to mid-FRET observations that are indistinguishable from those caused by a bona fide biomolecular conformation."

The sampling rate is 0.1 kHz (100 microsecond bin time) and all the rates are less than 0.5 s⁻¹. So, it doesn't look like too many transitions were missed due to their short dwell times to affect the extracted rates. It is also strange that the FRET efficiency difference between 1 and 3 is the largest, but k_{13} and k_{31} are least accurate. (see comment #5 below)

5. In Line 252, "Nevertheless, the steady-state flow was qualitatively well resolved by most tools."

I cannot agree with this statement. I have not followed the details of all the methods in this study, but some of the methods must have used a detailed balance condition in the kinetic schemes. In this case, the method is not appropriate for the steady state case like Fig. 3a. For example, ΔG value in Eq. (1) is 0 for the extracted parameters of Pomegranate in Fig. 3d. In addition, Tracy also runs HMM for the entire data initially, but later separates states using the Viterbi algorithm and determines the rates from the dwell time distribution. Therefore, it would be better than the first method (Pomegranate), but extracted rates cannot be correct. I suspect the deviations of kinetic parameters by other methods may result from this or similar reasons.

Since everyone knows what the simulated and experimental data are now, I suggest each group should go back to their analysis and comment on why their methods didn't work well if that is the case. This is a blind test, but in real cases, we know which system we work on. If we collect the data for a steady state system, then we would not apply for an equilibrium kinetic scheme. This explanation would better guide non-expert readers to choose appropriate models and methods.

Reviewer #3:

Remarks to the Author:

The manuscript of Gotz et al is a formidable work of comparison across multiple kinetic analysis tools commonly applied in the field of single-molecule FRET. A blind test of these different methodologies provides an important and honest benchmark for their effectiveness and their limitations. Overall, the emerging picture is of a convincing set of tools that enables resolving kinetics in multi-step processes. This work is of interest not only for the specialists of smFRET, but for the overall scientific community. Indeed, these results can offer orientation on which tools to embrace for using smFRET in resolving kinetics and how reliable they are for addressing the specific system at test. The discussion in the manuscript is balanced and very transparent. Importantly, this work clearly demonstrates a community effort to establish common tools and their validity, which I think is extremely important. Based on the high quality of the results and presentation, I recommend this work for publication with few minor suggestions, which I hope help improving accessibility of the text and discussion of the results.

Suggestions:

Line 76: The authors mention a typical time trace resolution of about 10 – 100 ms, but avalanche photodiodes instruments can provide picosecond information (depending on electronic) (e.g, the approach has been used to quantify also transition path times). The authors may want to expand on the time scales expected depending on camera or APD-based systems, which they already mentioned few lines above. Though not essential for the conclusions of the manuscript, this could be useful information for the non-expert readers in understanding the difference on the experimental methods used for gathering the data.

Box 1: The term BIC I am not sure the abbreviation is presented in the main text (it is described in the SI). In addition, the uncertainty sometimes is reported as a percentage without the confidence interval denomination: is there a reason?

Line 153: "FRETboard reported consistently very large uncertainties, which was solved in their latest software version". It is unclear by reading this sentence which version of FRETboard has been used in the analysis in the manuscript and, if an older version has been used, which data supports the second part of the statement.

It seems like information in Supplementary Figure 4 can be incorporated in Figure 4 to provide a direct sense of how the kinetic rates are reproduced. Indeed, while the cumulative distribution of dwell times looks quite similar, some of the kinetic rates are underestimated by all approaches. A comment on this point would help the reader putting the results in perspective.

Line 298: "while model selection remains a challenge" – the authors may want to clarify where the challenge is. Since most methods correctly capture the "model size" without a priori knowledge (line 284), the sentence seems to imply information regarding the connectivity of the states. Perhaps this can be made explicit.

In Figure 5 some of the points reported for the transfer efficiency determination do not match what is reported in Supplementary Figure 5. E.g., condition 2, tool 5, there is one single dot in Fig. 5 and 3 dots in S.Fig. 5. Can the authors explain why they choose to report the smallest model (as discussed in figure caption)? This seems confusing and sort of hiding details in an otherwise very transparent presentation of the results.

In addition, it could be useful to add – at least in the supplementary information and for the most successful methods – a graphical comparison of the determined rates.

Supplementary Table 1-4 could use a better identification of the quantities reported in each line and row.

One aspect that would deserve some discussion is the number of transitions (or trajectory length) as well as on the difference in the observed rates. All the experiments report about rates on a similar timescale. The exponential (or multi-exponential) decay of dwell-times strongly relies on

the available statistics of each transition, and it is unclear how it performs. A comparison of the methods on a truncated version of the same dataset (for one dataset, perhaps with one or two different truncations) could enable a further assessment on the similarities/dissimilarities of the approach. Similarly, a test of the kinetics for simulated data with order of magnitude different rates can provide a stronger stress test to the different methodologies. While these further experiments and analysis could add an additional layer of comparison, I understand if the authors would opt to address these points in a subsequent work given the complex effort of running such investigation as a blind test across many labs.

Reply to the Reviewers (original comments in black, our reply in blue font)

Reviewer #1 (Remarks to the Author):

smFRET provides the unique possibility to obtain structural and functional data from single labeled biomolecules. However, the interpretation of the noisy fluorescence and FRET time traces in the framework of kinetic models is not simple and usually, biochemical and other spectroscopy data provide kinetic models which are used as basis for the interpretation of smFRET data. In order to obtain the number of states and the kinetic constants directly from smFRET data several groups have developed procedures to analyse such data sets. These groups initiated the „kinetic software challenge“ to evaluate and to standardize experimental and analytical smFRET procedures.

In this joint approach each group used its own analysis tool (11 different tools, three of them were used under different conditions). They analyse the same FRET data and determined the number of states and the kinetic constants. They use the following data sets for this comparison: a kinetic 2-state equilibrium state system with simulated test data and with an experimental data set; a kinetic 3-state non-equilibrium state with simulated test data; a kinetic 4-state model with overlapping FRET-efficiencies with simulated kinetic heterogeneity, and finally, three experimental data sets with overlapping FRET-efficiencies.

The results obtained with the different tools are summarized in four instructive figures showing the FRET efficiency histograms with the assigned states and the rate constants with their uncertainties. The details of these comparisons are given in the Supplementary Information. Based on these results the authors formulated six general recommendations for users and developers of kinetic inference tools including a simple file format for dissemination of smFRET data.

This study is the first comparison of different kinetic analysis tools for inference of kinetic rate constants from smFRET trajectories. It describes the strengths and weaknesses of eleven software tools and it provides a reference standard for future developments in this field. This joint approach summarizes the knowledge of the participating groups and will help newcomers in this field to select the appropriate tool for their data analysis.

The manuscript is clearly written, data files are provided in the supporting material, the results of the data analysis are presented in very instructive figures, data analysis and statistics are described in the different tools. This work provides reference standards for the analysis of smFRET trajectories and is benchmark for further developments.

I strongly recommend publication.

We thank the reviewer for this very positive and detailed assessment of our work, for recognizing the importance of our study providing a reference standard for future developments, and for the recommendation to publish our work in Nature Communications.

Minor remarks:

Unfortunately, there is no common standard for the uncertainties in the different tools what complicates the comparison of these results.

We agree with the Reviewer: so far, the comparability of uncertainty estimates has been hampered by the lack of a common standard. This is an important point that we mentioned in the Discussion section. To further stress this aspect, and encourage the unification of the uncertainty measures in the future, we now added a statement on this to the 'general recommendations' in Box 2 (iv). We also updated three entries in Box 1 to clarify the provided uncertainty estimates.

A description of each tool is given in the Supplementary Methods. These descriptions differ greatly in detail, some using half a page, some 5 pages. A consistent description would be very helpful as overview for non-experts in this field.

We thank the Reviewer for providing this reader feedback. To unify the Supplementary Methods, we now introduced a common structure for all tool descriptions, with the same headings: A) Overview, B) Workflow, C) Miscellaneous.

Reference 37 is wrong. It refers to the F1- ATPase where a three stepped rotation of an antibody bound to the gamma subunit driven by ATP hydrolysis was observed. The first observation of a three stepped rotation driven by a chemical gradient in ATP synthesis direction and in opposite direction during ATP hydrolysis was carried with the smFRET technique using a membrane integrated FOF1 ATP synthase. Diez et al. (2004) Nature Structural & Molecular Biology 11, 135-141

We thank the Reviewer for this correction, which we incorporated as suggested.

Reviewer #2 (Remarks to the Author):

This manuscript reports the results of a recent blind test study of the kinetic analysis of single molecule fluorescence data. Various simulated and experimental data were analyzed using 14 different methods by expert groups and the performances were compared. In addition to the analysis methods introduced and compared in this study, there must be other methods and variations, some of which are not publicly accessible. Various photon-by-photon analysis methods are not included here, either. However it would not be possible to cover everything, and I believe this is a timely and appropriate study that should be published. The blind data set was designed well, including simulation (ground truth available) and real experimental data with a relatively easy, two-state system and more complex three- and four- state systems (both equilibrium and non-equilibrium systems). Overall, the analysis results are well summarized and presented. However, there are several models that cannot be analyzed well by some of the methods. I don't think this is not because those methods are poor or have problems, but because those methods are not compatible with the data model type. The authors could not know this before the manuscript was written. So I suggest adding more comments and explanations for the cases that the methods don't work well. See the comments below for more detailed suggestions and other minor issues.

We thank the Reviewer for appreciating the importance, timeliness and appropriate design of our study, and for recommending its publication in Nature Communication. We appreciate the suggestions and minor issues raised, based on which we further improved our manuscript.

1. Page 6, 2-state simulated data:

Usually, the errors of the rate coefficients are larger than those of the FRET efficiencies. However, the maximum deviation of the FRET efficiency 17% is larger than that of the rate 12% although the deviation of the FRET efficiencies seems very small in Fig. 2C and F. Therefore, I am not sure if comparing the maximum variation is right. What is the average deviation of the FRET efficiency? (This is 5% for the rate.) This could be a better comparison.

We thank the Reviewer for this comment. The average deviation of the FRET efficiency is 1% in Figure 2c, which we now specify on line 140 of the revised manuscript. So, as expected by the Reviewer, it is indeed smaller than the average deviation of the rate constants.

2. In Line 152. "FRETboard reported consistently very large uncertainties, which was solved in their latest software version 0.0.3 (Ref33). Step finding initially found erroneously large uncertainties that have been corrected in the latest software version."

In this case, it may be good to include the updated uncertainties by the latest versions in a note.

We thank the Reviewer for mentioning this. We now added the updated uncertainties by FRETboard and StepFinding to the Supplementary Datafiles, which we mention on lines 154,155 of the revised manuscript and in the caption of Figure 2.

3. In Fig. 2, the dwell times and number of transitions within 100 s time window of the simulated data (Fig. 2b) and experimental data (Fig. 2e) are comparable. However, the extracted rate constants in Fig. 2d and g differ by more than a factor of 10. It looks like the y-axis in Fig. 2g was mislabeled.

We thank the Reviewer for mentioning this misunderstanding. As specified in Figures 2b,e, the axis labelling is in 'data points' – which is the relevant unit for comparing the raw data. In addition, we provided a scale bar that states the separate units of time. We now positioned this scale bar more prominently on the left, to prevent any misunderstandings. Additionally, the time information was already stated as the sampling rate in all captions of Figures 2-5. We now also state the reciprocal – the time per datapoint – in all those captions.

4. In line 235, “Most tools systematically underestimated k_{13} and k_{31} and overestimated all other rate constants. This may be attributed to the inevitable effect of time discretization and related intensity averaging: when a transition between the high- and low-FRET states happens during a time bin, time-weighted averaging (camera blurring) of the FRET efficiencies occurs, leading, in some cases, to mid-FRET observations that are indistinguishable from those caused by a bona fide biomolecular conformation.”

The sampling rate is 0.1 kHz (100 microsecond bin time) and all the rates are less than 0.5 s⁻¹. So, It doesn't look like too many transitions were missed due to their short dwell times to affect the extracted rates. It is also strange that the FRET efficiency difference between 1 and 3 is the largest, but k_{13} and k_{31} are least accurate. (see comment #5 below)

The Reviewer refers here to the data in Fig. 3, which has a sampling rate of 10Hz (as stated in the figure caption), resulting in 100 milliseconds bin time – not 100 *microseconds* as mentioned in the Reviewers comment. The time resolution is thus just a factor ~10 faster than the fastest observed dynamics (in contrast to a factor of ~10'000 as incorrectly implied in the Reviewer's comment). Therefore, time averaging is indeed relevant here. And k_{13} and k_{31} are the least accurate rates, because they are the fastest rates. This is the case *despite* the largest FRET E difference between states 1 and 3, because of the time averaging that blurs short dwells and state transitions (illustrated here below), which are sometimes misinterpreted as mid-FRET states. We clarified our description of the effect on line 239-241 of the revised manuscript. Plus, in addition to the sampling rate, we now also state its reciprocal, the 'time per datapoint' in all captions, to prevent confusion.

5. In Line 252, “Nevertheless, the steady-state flow was qualitatively well resolved by most tools.”

I cannot agree with this statement. I have not followed the details of all the methods in this study, but some of the methods must have used a detailed balance condition in the kinetic schemes. In this case, the method is not appropriate for the steady state case like Fig. 3a. For example, ΔG value in Eq. (1) is 0 for the extracted parameters of Pomegranate in Fig. 3d. In addition, Tracy also runs HMM for the entire data initially, but later separates states using the Viterbi algorithm and determines the rates from the dwell time distribution. Therefore, it would be better than the first method (Pomegranate), but extracted rates cannot be correct. I suspect the deviations of kinetic parameters by other methods may result from this or similar reasons.

Since everyone knows what the simulated and experimental data are now, I suggest each group should go back to their analysis and comment on why their methods didn't work well if that is the case. This is a blind test, but in real cases, we know which system we work on. If we collect the data for a steady state system, then we would not apply for an equilibrium kinetic scheme. This explanation would better guide non-expert readers to choose appropriate models and methods.

The Reviewer is right that IF the detailed balance condition is enforced, steady-state flow can – per definition – not be inferred. However, the Reviewer's assumption that '*some of the methods must have used a detailed balance condition in the kinetic schemes*' is not correct: none of the considered tools used a detailed balance constraint to infer ΔG . Notably, our manuscript does not state nor imply that detailed balance conditions were used during rate inference. We thank the Reviewer for raising this point, which we clarify now explicitly on lines 252-254 of the revised manuscript.

Regarding the observed inaccuracies in net flow: those were reanalyzed as suggested by the reviewer and they were found to be linked to the time-averaging effect leading to misallocated transitions, as described above in our reply to point 4, and as stated in the manuscript on lines 248/249. As an example: for the Pomegranate analysis mentioned by the Reviewer, this averaging effect led to the imprecise rate constants shown in Fig. 3d, which then results in the underestimated net flow (slightly negative).

(A technical note on HMM which was mentioned by the Reviewer: detailed balance is not automatically enforced in HMMs. Specific constraints are necessary to enforce detailed balance in HMM. See e.g.: Greenfeld, Pavlichin, Mabuchi, Herschlag *PLoS ONE*, 7(2):e30024, 2012. doi:10.1371/journal.pone.0030024 .)

Reviewer #3 (Remarks to the Author):

The manuscript of Gotz et al is a formidable work of comparison across multiple kinetic analysis tools commonly applied in the field of single-molecule FRET. A blind test of these different methodologies provides an important and honest benchmark for their effectiveness and their limitations. Overall, the emerging picture is of a convincing set of tools that enables resolving kinetics in multi-step processes. This work is of interest not only for the specialists of smFRET, but for the overall scientific community. Indeed, these results can offer orientation on which tools to embrace for using smFRET in resolving kinetics and how reliable they are for addressing the specific system at test. The discussion in the manuscript is balanced and very transparent. Importantly, this work clearly demonstrates a community effort to establish common tools and their validity, which I think is extremely important. Based on the high quality of the results and presentation, I recommend this work for publication with few minor suggestions, which I hope help improving accessibility of the text and discussion of the results.

We thank the Reviewer for the enthusiastic appraisal of our work, for deeming it an important and honest benchmark of broad scientific interest, and for the recommendation to publish it in Nature Communication.

Suggestions:

Line 76: The authors mention a typical time trace resolution of about 10 – 100 ms, but avalanche photodiodes instruments can provide picosecond information (depending on electronic) (e.g, the approach has been used to quantify also transition path times). The authors may want to expand on the time scales expected depending on camera or APD-based systems, which they already mentioned few lines above. Though not essential for the conclusions of the manuscript, this could be useful information for the non-expert readers in understanding the difference on the experimental methods used for gathering the data.

We thank the Reviewer for this helpful suggestion. Beyond stating the currently *typical* time trace resolution, we now expanded the information on accessible timescales on lines 76-77 of the revised manuscript, as suggested.

Box 1: The term BIC I am not sure the abbreviation is presented in the main text (it is described in the SI). In addition, the uncertainty sometimes is reported as a percentage without the confidence interval denomination: is there a reason?

We thank the Reviewer for mentioning this oversight, which we regret. We now clarified the acronym BIC and corrected the mentioned uncertainty descriptors in Box 1.

Line 153: “FRETboard reported consistently very large uncertainties, which was solved in their latest software version”. It is unclear by reading this sentence which version of FRETboard has been used in the analysis in the manuscript and, if an older version has been used, which data supports the second part of the statement.

We thank the Reviewer for raising this point. We now clarified the version numbers on lines 153-155 (also in response to Reviewer 2).

It seems like information in Supplementary Figure 4 can be incorporated in Figure 4 to provide a direct sense of how the kinetic rates are reproduced. Indeed, while the cumulative distribution of dwell times looks quite similar, some of the kinetic rates are underestimated by all approaches. A comment on this point would help the reader putting the results in perspective.

We followed the Reviewer’s suggestion and incorporated Supplementary Figure 4 into Figure 4e of the main text, and we added a comment on lines 301-304, as suggested. In addition, we corrected a small misrepresentation in the cartoon in Fig. 4a, where the thin and thick arrows (indicating slow and fast rates, respectively) were previously inverted.

Line 298: “while model selection remains a challenge” – the authors may want to clarify where the challenge is. Since most methods correctly capture the “model size” without a priori knowledge (line 284), the sentence seems to imply information regarding the connectivity of the states. Perhaps this can be made explicit.

The Reviewer is right: the connectivity of states can still be challenging to infer. We now made this more specific by rephrasing lines 304ff of the revised manuscript.

In Figure 5 some of the points reported for the transfer efficiency determination do not match what is reported in Supplementary Figure 5. E.g., condition 2, tool 5, there is one single dot in Fig. 5 and 3 dots in S.Fig. 5. Can the authors explain why they choose to report the smallest model (as discussed in figure caption)? This seems confusing and sort of hiding details in an otherwise very transparent presentation of the results.

As stated in the caption of Figure 5, and as also acknowledged by the Reviewer: “*For clarity, only the smallest reported model is shown for each analysis tool, up to a maximum of four FRET states. All inferred FRET efficiencies are shown in Supplementary Figure 5 (...).*” So clearly, nothing was hidden. The choice for the smallest model is in line with parsimony principles: the lowest number of parameters needed to describe the data well is generally preferred (cf. Occam’s razor).

We appreciate that the Reviewer recognizes the ‘*very transparent presentation of the results*’ because this was our absolute goal – also in Figure 5/Supplementary Figure 5, which we think are clear. If there are specific suggestions on how to further improve them, we are happy to do so, while we also note that none of the other reviewers found these figures unclear.

In addition, it could be useful to add – at least in the supplementary information and for the most successful methods – a graphical comparison of the determined rates.

If all tools were to infer the same connectivity of states, the suggested ‘*graphical comparison of the determined rates*’ would have been ideal and our preference. However, in reality, the state connectivity differs (also among tools with the same number of states) and the comparison of the resulting *conceptually* different rates is not meaningful, and hence it would be misleading. In addition, for this experimental data, no ground truth is available for comparison. This is why we turned to the re-simulated CDFs, which allows for a meaningful comparison despite varied state connectivities. This aspect was described in the discussion of Figure 5c,f,i, and we now edited our description on lines 349 and 352 of the revised manuscript to further improve clarity.

Supplementary Table 1-4 could use a better identification of the quantities reported in each line and row.

We thank the Reviewer for this suggestion. We now label each provided value individually, throughout all Supplementary Tables 1-4.

One aspect that would deserve some discussion is the number of transitions (or trajectory length) as well as on the difference in the observed rates. All the experiments report about rates on a similar timescale. The exponential (or multi-exponential) decay of dwell-times strongly relies on the available statistics of each transition, and it is unclear how it performs. A comparison of the methods on a truncated version of the same dataset (for one dataset, perhaps with one or two different truncations) could enable a further assessment on the similarities/dissimilarities of the approach. Similarly, a test of the kinetics for simulated data with order of magnitude different rates can provide a stronger stress test to the different methodologies. While these further experiments and analysis could add an additional layer of comparison, I understand if the authors would opt to address these points in a subsequent work given the complex effort of running such investigation as a blind test across many labs.

We thank the Reviewer for sharing these forward-looking suggestions. We agree that the number of transitions per trace (i.e. the relation of biomolecular dynamics vs. bleach rate) is an interesting aspect for future investigation, same as the described ‘*data greediness*’ of the various approaches, and how they cope with slow and fast rates. We gladly added these aspects to our outlook section (lines 431-434), and keep the suggestion in mind for the next challenge round.

Reviewers' Comments:

Reviewer #1:

Remarks to the Author:

All my concerns are treated appropriately. I recommend publication of the revised manuscript.

Reviewer #2:

Remarks to the Author:

The authors have addressed all my questions and concerns. I recommend the publication of this work.

Reviewer #3:

Remarks to the Author:

The new version of the manuscript addresses all my concerns. I thank the authors for addressing my concerns.

Reply to reviewers (reviewer comments in black, our reply in blue)

Reviewer #1 (Remarks to the Author):

All my concerns are treated appropriately. I recommend publication of the revised manuscript.

Reviewer #2 (Remarks to the Author):

The authors have addressed all my questions and concerns. I recommend the publication of this work.

Reviewer #3 (Remarks to the Author):

The new version of the manuscript addresses all my concerns. I thank the authors for addressing my concerns.

We thank all three reviewers for their careful reading of our manuscript, for their thoughtful comments, and for their recommendation to publish the revised manuscript.